# Inhibitory Effect of *Antidesma bunius* Fruit Extract on Carbohydrate Digestive Enzymes Activity and Protein Glycation In Vitro

**DOI:** 10.3390/antiox10010032

**Published:** 2020-12-30

**Authors:** Pattamaporn Aksornchu, Netima Chamnansilpa, Sirichai Adisakwattana, Thavaree Thilavech, Charoonsri Choosak, Marisa Marnpae, Kittana Mäkynen, Winai Dahlan, Sathaporn Ngamukote

**Affiliations:** 1Phytochemical and Functional Food Research Unit for Clinical Nutrition, Department of Nutrition and Dietetics, Faculty of Allied Health Sciences, Chulalongkorn University, Bangkok 10330, Thailand; paksornchu@gmail.com (P.A.); m.netti7@gmail.com (N.C.); sirichai.a@chula.ac.th (S.A.); charoonsri.c@gmail.com (C.C.); mmarisa.hsc@gmail.com (M.M.); kittana.m@chula.ac.th (K.M.); 2Department of Food Chemistry, Faculty of Pharmacy, Mahidol University, Bangkok 10400, Thailand; thavaree.thi@mahidol.ac.th; 3The Halal Science Center, Chulalongkorn University, Bangkok 10330, Thailand; winai.hsc@gmail.com

**Keywords:** α-amylase, α-glucosidase, *Antidesma bunius*, protein glycation, anthocyanins, antioxidant

## Abstract

*Antidesma bunius* (L.) spreng (Mamao) is widely distributed in Northeastern Thailand. *Antidesma bunius* has been reported to contain anthocyanins, which possess antioxidant and antihypertensive actions. However, the antidiabetic and antiglycation activity of *Antidesma bunius* fruit extract has not yet been reported. In this study, we investigated the inhibitory activity of anthocyanin-enriched fraction of *Antidesma bunius* fruit extract (ABE) against pancreatic α-amylase, intestinal α-glucosidase (maltase and sucrase), protein glycation, as well as antioxidant activity. A liquid chromatography-tandem mass spectrometry (LC-MS/MS) chromatogram revealed that ABE contained phytochemical compounds such as cyanidin-3-glucoside, delphinidin-3-glucoside, ellagic acid, and myricetin-3-galactoside. ABE inhibited intestinal maltase and sucrase activity with the IC_50_ values of 0.76 ± 0.02 mg/mL and 1.33 ± 0.03 mg/mL, respectively. Furthermore, ABE (0.25 mg/mL) reduced the formation of fluorescent AGEs and the level of N^ε^-carboxymethyllysine (N^ε^-CML) in fructose and glucose-induced protein glycation during four weeks of incubation. During the glycation process, the protein carbonyl and β-amyloid cross structure were decreased by ABE (0.25 mg/mL). In addition, ABE exhibited antioxidant activity through DPPH radical scavenging activity and Trolox equivalent antioxidant capacity (TEAC) with the IC_50_ values 15.84 ± 0.06 µg/mL and 166.1 ± 2.40 µg/mL, respectively. Meanwhile, ferric reducing antioxidant power (FRAP) showed an EC_50_ value of 182.22 ± 0.64 µg/mL. The findings suggest that ABE may be a promising agent for inhibiting carbohydrate digestive enzyme activity, reducing monosaccharide-induced protein glycation, and antioxidant activity.

## 1. Introduction

Diabetes is a group of abnormal carbohydrate, protein, and lipid metabolisms that contributes to chronically elevated blood glucose levels, known as hyperglycemia [1,2]. Long-term carbohydrate consumption, including the type and amount of dietary carbohydrate, is the leading risk factor for the induction of hyperglycemia, causing insulin resistance by acting on targeting cells such as adipose tissue and muscle cells [3,4,5,6]. After meal intake, starch is digested by pancreatic α-amylase to produce oligosaccharides and disaccharides and further hydrolyzed by intestinal α-glucosidase to yield absorbable monosaccharides [7,8]. Consequently, monosaccharides’ absorption into the bloodstream occurs in the small intestine through glucose transporters, increasing postprandial blood glucose [1]. In diabetic patients, uncontrolled hyperglycemia leads to the onset and progression of diabetic vascular complications such as retinopathy, nephropathy, neuropathy, and cardiovascular diseases [9]. The important pathway involved in the pathogenesis of diabetic complications is the formation of advanced glycation endproducts (AGEs), resulting from non-enzymatic glycation between the carbonyl group of reducing sugar and the free amino group of protein [10]. The scientific evidence suggests that the binding of AGEs and receptors for AGEs (RAGEs) triggers the intracellular signaling involving the activation of reactive oxygen species (ROS) generation, inflammatory cytokine production, macrophage-platelet interaction, and thrombotic reaction, leading to vascular damage [11]. The overproduction of ROS during the glycation process results in the deterioration of biological molecules such as protein, lipid, and DNA, contributing to the pathogenesis of several diseases such as Alzheimer’s diseases, cardiovascular disease, and cancer [9].

The previous study revealed that the inhibition of carbohydrate digestive enzymes could delay the monosaccharide absorption into the bloodstream, resulting in the decreased postprandial blood glucose concentration [12]. Moreover, achieving glycemic control or reducing hyperglycemia markedly decreases the level of glycated protein (HbA1c), an important risk indicator for long-term diabetic complications [13]. In clinical practice, acarbose, α-glucosidase inhibitor, has been used to manage postprandial glucose level in diabetic patients [14,15]. However, aminoguanidine, a promising antiglycation agent, was terminated in clinical trials with serious side effects such as flu-like symptoms, anemia, vasculitis, myocardial infarction, congestive heart failure, and arterial fibrillation [9,10,16]. 

Nowadays, the inhibition of carbohydrate digestion and the prevention of AGE formation has been of great interest to lower postprandial hyperglycemia and decrease protein glycation, especially in plant-derived products containing phytochemicals [17,18,19]. For example, anthocyanin-rich berries exhibited antioxidants, anti-glycation, antidiabetic, including pancreatic α-amylase and intestinal α-glucosidase inhibitions in in vitro studies [20,21]. Berries, including black raspberry, blueberry, blackberry, have also been reported to inhibit fructose-induced protein glycation and aggregation [22]. Moreover, the extract of *Clitoria ternatea* flower inhibited the formation of AGEs and prevented oxidative damage to bovine serum albumin (BSA) [23].

*Antidesma bunius* (L.) Spreng, Mamao, a berry of the family Euphorbiaceae, is locally cultivated and distributed in Northeastern Thailand. Interestingly, the extract of *Antidesma bunius* fruit contains polyphenols, including gallic acid, epicatechin, catechin, and cyanidin-3-glucoside, which demonstrate antioxidant activity in various models [24]. It also exhibited antihypertensive activity by scavenging superoxide radicals, inhibiting the formation of lipid and protein oxidation, and increasing expression of eNOS in N^ω^-nitro-L-arginine methyl ester-induced hypertensive rats [25]. In high fat diet-induced rats, the extract of *Antidesma bunius* fruit suppressed the gene expression of lipogenic enzymes and reduced the level of triglyceride and inflammation [26,27]. Although the extract of *Antidesma bunius* fruit shows various biological properties, its inhibitory effect against carbohydrate digestive enzymes and protein glycation remains unknown. Therefore, this study aimed to investigate the inhibitory activity of anthocyanin-enriched fraction of *Antidesma bunius* fruit extract (ABE) on pancreatic α-amylase and intestinal α-glucosidase. Moreover, this study also examined ABE’s inhibitory effect on fructose/glucose-induced protein glycation, oxidative protein damage, and protein aggregation. Finally, the phytochemical composition and antioxidant activity of ABE was evaluated.

## 2. Materials and Methods

### 2.1. Chemicals

Folin-Cioclateu’s phenol reagent, gallic acid, porcine pancreatic α-amylase (Type VI-B), rat intestinal acetone powder, 3,5-dinitrosalicylic acid, bovine serum albumin (BSA, fraction V), aminoguanidine hydrochloride (AG), guanidine hydrochloride, thioflavin T, 2,2-diphenyl-1-picrylhydrazyl (DPPH), 2,2′-Azino-bis(3-ethylbenzothiazoline-6-sulfonic acid) diammonium salt (ABTS), 6-hydroxy-2,5,7,8-tetramethylchromane-2-carboxylic acid (Trolox), and 2,4,6-tris(2-pyridyl)-s-triazine (TPTZ), delphinidin-3-glucoside (D3G) were purchased from Sigma-Aldrich Co. (St. Louis, MO, USA). 2,4-dinitrophenylhydrazine (DNPH), D-maltose, D-fructose, D-glucose, sucrose, and iron (II) sulfate were purchased from Ajax Finechem (Taren Point, Australia). Trichloroacetic acid (TCA) was purchased from Merck (Darmstadt, FR, Germany). A glucose liquid color kit was purchased from HUMAN Gesellschaft für Biochemica und Diagnostica mbH (Wiesbaden, Germany). Acarbose was purchased from Bayer AG Pharmaceutical (Berlin, Germany). Cyanidin-3-O-glucoside (C3G) was obtained from PhytoLab GmbH & Co. KG (Vestenbergsgreuth, Germany). All other chemicals used in this study were analytical grade.

### 2.2. Plant Materials

The dark-purple color fruit (mature ripened stage) of *Antidesma bunius* was collected during August–September 2017 from Sakon Nakhon province, Northeastern Thailand. The plant was authenticated by a taxonomist Parinyanoot Klinratana at Professor Kasin Suvatabhandhu Herbarium, Department of Botany, Faculty of Science, Chulalongkorn University, Thailand. The voucher specimen was 015866 (BCU).

### 2.3. Sample Preparation and Extraction

The fruit extraction was performed according to a previous method with a slight modification [28]. The fruit (10 kg) was deseeded and blended with distilled water (20 L). The sample was filtered through a sieve cloth and centrifuged at 3500 rpm for 5 min at 4 °C. The solution was lyophilized using a freeze-dryer. The freeze-dried powder (1 g) was dissolved in distilled water (20 mL). The anthocyanin-rich fraction was purified using C_18_ solid-phase extraction (Strata-C18E, Phenomenex Inc., Torrance, CA, USA) preconditioned with 0.2% formic acid in acetonitrile (6 mL) and then pre-equilibrated with 0.2% formic acid in water (6 mL). Consequently, the sample (6 mL) was loaded into an extraction tube, washed with 0.2% formic acid in water (6 mL) and water (12 mL). After that, acetonitrile (80% *v/v* in water) was used to elute the anthocyanin-rich fraction. This fraction was evaporated at 50 °C (BUCHI ROTAVAPOR, Flawil, Switzerland) to remove the solvent. The anthocyanin-rich *Antidesma bunius* fruit extract (ABE) was kept at −20 °C until use. In the experiment, ABE was dissolved in distilled water.

### 2.4. Determination of Total Phenolic and Anthocyanin Content

The total phenolic content of ABE was determined using the Folin–Ciocalteau method [14]. In brief, 1 mg/mL ABE (50 µL) was mixed with 50 µL Folin-Ciocalteu reagent (10-fold dilution with distilled water) and 50 µL Na_2_CO_3_ (10% *w/v*). After 30 min of incubation in the dark, the absorbance was measured at 760 nm. The phenolic content was expressed as mg gallic acid equivalent (GAE)/g extract.

The total anthocyanin content of ABE was measured by the pH differential method [29]. The extract (1 mg/mL; 250 µL) was mixed with 750 µL of two different buffer systems, 0.025 M potassium chloride (pH 1.0) and 0.4 M sodium acetate (pH 4.5). All sample solutions were incubated at room temperature for 20 min in the dark and monitored the absorbance at a wavelength of 520 nm and 700 nm. The absorbance of the solution was calculated by the following formula: A = (A520-A700)pH1.0 − (A520-A700)pH4.5. The anthocyanin content was obtained from
(1)$$\mathrm{Anthocyanin}\;\mathrm{content}\;(\mathrm{mg/L})=\;\frac{A\;\times \;\mathrm{MW}\;\times \;\mathrm{DF}\;\times \;1000}{\varepsilon \;\times \;\lambda \;\times \;c}$$
where A is the absorbance of sample, MW is the molecular weight of cyanidin-3-glucoside (C3G), ε is a molar extinction coefficient of C3G as 26,900 L·mol^−1^ cm^−1^, λ is the path length (cm), c is the sample concentration (mg/L). Anthocyanin content was expressed as mg C3G equivalent/g extract.

### 2.5. Identification of Phytochemical Compounds

The phytochemical compounds in ABE were characterized using liquid chromatography-mass spectrometry (LC-MS/MS) according to a previously published method with a minor modification [30]. The HPLC system was equipped with a 6545 Quadrupole-time of flight (TOF) Mass spectrometer (Agilent Technologies, Santa Clara, CA, USA). The phenolic compounds were separated using the Luna C_18_ column (150 × 2.0 mm, 3 µm, Phenomenex Inc., Torrance, CA, USA). The mobile phase consisted of (A) water: formic acid: acetonitrile (95:2:3 *v/v/v*); and (B) water: formic acid: acetonitrile (48:2:50 *v/v/v*). The flow rate was set at 0.20 mL/min. The gradient program was 5% (B), 25% (B) in 50 min, and 5% (B) in 5 min. The injection volume was 5 µL. The phenolic compounds were identified in negative and positive modes with a mass range of *m/z* 100 to 3000. The mass spectrophotometer (MS) condition had the following parameters: nitrogen gas temperature at 325 °C, gas flow at 11 L/min, nebulizer gas at 35 psi, sheath gas temperature at 350 °C, sheath gas flow at 11 L/min, capillary at 3500 V, and fragmentor voltage at 125 V. The interpretation of MS/MS data were carried out using Respect, GNPS, and MassBank mass spectral libraries. The identification score of 80% was selected for mass verification.

### 2.6. Quantification of Individual Anthocyanin

According to a previous method with modification, the individual anthocyanin in ABE was quantified using LC-MS/MS system [31,32]. The high-performance liquid chromatography (HPLC) system (Agilent 1290, Agilent Technologies, Santa Clara, CA, USA) consisted of a binary pump and autosampler with a reverse-phase C-18 Inertsil ODS-2 column (250 × 4.6 mm, 5 µm, 150 A°, Techno Quartz Inc., Tokyo, Japan). The mobile phase was formic acid: water (10:90 *v/v*; mobile phase A) and formic acid: water: acetonitrile: methanol (10:40:22.5:22.5 *v/v/v/v*; mobile phase B). The anthocyanin was separated by a linear gradient following: 15–20% (B) in 5 min, 20–27% (B) in 35 min, 27–65% (B) in 45 min, 65–100% (B) in 50 min and then back to 15% (B) until 60 min at a flow rate of 0.6 mL/min. The injection volume of the sample was 5 µL. The anthocyanin was quantified using a mass spectrophotometer with electrospray ionization (ABSciex QTRAP 5500, Sciex, Framingham, MA, USA). The anthocyanin content was determined by the multiple reaction monitoring-enhanced product ion mode (MRM-EPI) under condition was set as follows: ion spray voltage 5.5 kV, source temperature 500 °C, curtain gas 25 psi, collision energy 20 eV. Data were analyzed using “ABSciex analyst” software (Sciex, Framingham, MA, USA). Anthocyanins standard was cyanidin-3-glucoside (C3G) and delphinidin-3-glucoside (D3G).

### 2.7. The Inhibition of Carbohydrate Digestive Enzymes

The inhibition of pancreatic α-amylase was performed according to a previous report [14]. In brief, the ABE at a concentration of 15–60 mg/mL (20 µL) and 4 g/l starch solution (75 µL) were mixed in a 0.1 M phosphate buffer, pH 6.9 (130 µL). After the addition of 75 µL pancreatic α-amylase (three units/mL), the mixture was incubated at 37 °C for 10 min. Then, 1% dinitrosalicylic acid (250 µL) was added to the mixture and heated for 10 min to stop the enzyme activity. The mixture was incubated with 40% potassium sodium tartrate (250 µL) and kept at room temperature. The absorbance was monitored at 540 nm. In this study, acarbose was used as a positive control. The inhibitory activity was expressed as a percentage of inhibition.
(2)$$\%\;\mathrm{inhibition}\;=\;\left[\frac{{\mathrm{Abs}}_{\mathrm{control}}-{\mathrm{Abs}}_{\mathrm{sample}}}{{\mathrm{Abs}}_{\mathrm{control}}}\right]\times 100$$

Abs_control_ was absorbance without ABE or acarbose, and Abs_sample_ was the absorbance of ABE or acarbose.

The inhibition of intestinal α-glucosidase was determined following a previously published method [14]. The rat intestinal acetone powder was dissolved in 0.9% NaCl solution at a concentration of 100 mg/3 mL and centrifuged at 12,000× *g* for 30 min at 4 °C. The supernatant was collected for further analysis. In the assay of maltase and sucrase activity, ABE (10 µL) was mixed with 86 mM maltose (30 µL) or 400 mM sucrose (40 µL) in 0.1 M phosphate buffer, pH 6.9, respectively. The reaction was incubated at 37 °C for 30 min (maltase assay) or 60 min (sucrase assay). Then, the reaction was heated at 100 °C for 10 min. The release of glucose was determined by the glucose oxidase method. The absorbance was measured at 500 nm. In this study, acarbose (0.16–5 µg/mL) was used as a positive control. The results were expressed as the half-maximal inhibitory concentration (IC_50_) values (the concentration required to the percentage of inhibition, with 50% of the enzyme activity) that were calculated using the curve of logarithmic regression.

### 2.8. Protein Glycation

The protein glycation assay was performed following a published method [33]. Briefly, 700 µL of BSA (10 mg/mL final concentration) was mixed with 560 µL of fructose or glucose (0.5 M final concentration) in 0.1 M phosphate buffer saline (PBS, pH 7.4) containing 0.02% sodium azide. The solution was mixed with 140 µL of ABE in 0.1 M PBS (0.025–0.25 mg/mL final concentration) or aminoguanidine (0.25 mg/mL final concentration). The solution was incubated at 37 °C in the dark for four weeks. The fluorescent intensity of AGE formation was measured weekly using a spectrofluorometer (Fluoroskan Ascent, Thermo Fisher Scientific, Vantaa, Finland) at excitation and emission wavelength 355 nm and 460 nm, respectively. The following equation calculated the percentage inhibition
Inhibition of fluorescent AGEs (%) = ((F_C_ − F_CB_) − (F_S_ − F_SB_)/(F_C_ − F_CB_)) × 100(3)
where F_C_ and F_CB_ were the fluorescent intensity of control with monosaccharide and blank of control without monosaccharide. F_S_ and F_SB_ were the fluorescent intensity of the sample with monosaccharide and blank of the sample without monosaccharide.

The level of N^ε^-carboxymethyllysine (N^ε^-CML), a non-fluorescent AGEs, was analyzed by enzyme-linked immunosorbent assay (ELISA) kit following the manufacturer’s protocol (Cyclex Co., Ltd., Nagano, Japan). The absorbance was measured at 450 nm. The level of N^ε^-CML was calculated by the CML-human serum albumin (HSA) standard curve.

### 2.9. Determination of Protein Carbonyl Content

The content of protein carbonyl was determined using a 2,4-dinitrophenylhydrazine (DNPH) assay according to a previous report [33]. Briefly, the glycated BSA (100 µL) was mixed with 10 mM DNPH in 2.5 M HCl (400 µL) and incubated for 60 min in the dark. After incubation, the protein was precipitated by 20% (*w/v*) trichloroacetic acid (500 µL), kept on ice for 5 min, and centrifuged at 10,000× *g* for 10 min at 4 °C. The protein pellet was washed with methanol: ethyl acetate (1:1 *v/v*) three times and dissolved in 250 µL 6 M guanidine hydrochloride. The absorbance was read at 370 nm. The carbonyl content was calculated using the molar extinction coefficient of 22,000 (DNPH). The results were expressed as nmol carbonyl/mg protein.

### 2.10. Determination of Protein Aggregation

The level of β-amyloid cross structure was analyzed by thioflavin T, according to a previous study [33]. The glycated BSA (50 µL) was mixed with 64 µM thioflavin T (50 µL) in 0.1 M PBS, pH 7.4. The mixture was incubated at room temperature for 60 min. The β-amyloid cross structure level was measured using a spectrofluorometer at excitation and emission wavelength with 435 nm and 485 nm, respectively.

### 2.11. 2,2-Diphenyl-1-picrylhydrazyl (DPPH) Scavenging Activity

2,2-Diphenyl-1-picrylhydrazyl (DPPH) radical scavenging activity was determined following the previous protocol [34]. In brief, 100 µL of ABE (10–60 µg/mL) was mixed with 100 µL of 0.2 mM DPPH in ethanol and incubated for 30 min at room temperature. The absorbance was measured at 515 nm. Ascorbic acid (5–30 µg/mL) was used as a positive control. The IC_50_ value was calculated using the curve of the percentage of DPPH scavenging activity corresponding to the various sample concentrations.

### 2.12. Trolox Equivalent Antioxidant Capacity (TEAC)

Trolox Equivalent Antioxidant Capacity (TEAC) was analyzed according to a previously published method [35]. In brief, 2.45 mM potassium sulfate was mixed with 7 mM ABTS at a ratio of 1:1 and incubated at room temperature for at least 16 h in the dark to generate ABTS^+^ radicals. After incubation, the ABTS^+^ solution was diluted with 0.1 M PBS, pH 7.4, to obtain the absorbance at 0.9–1.0 before analysis. ABE, at a concentration of 0.02–2 mg/mL (20 µL,) was incubated with 180 µL ABTS^.+^ solution and incubated at room temperature for 6 min. Trolox (0.06–1 mg/mL) was used as a positive control. The absorbance was measured at 734 nm. The IC_50_ value was calculated using the curve of the percentage of ABTS^+^ scavenging activity corresponding to the various sample concentrations.

### 2.13. Ferric Reducing Antioxidant Power (FRAP)

The Ferric Reducing Antioxidant Power (FRAP) value was determined according to a previous study [24]. The FRAP reagent was prepared by mixing 10 mM 2,4,6-Tris(2-pyridyl)-s-triazine (TPTZ) in 40 mM HCl with 20 mM FeCl_3_ and 300 mM acetate buffer (1:1:10 *v/v/v*). The FRAP reagent was incubated at 37 °C before analysis. The ABE at a concentration of 0.02–1 mg/mL (20 µL) was mixed with 180 µL of FRAP reagent and incubated at 37 °C for 30 min in the dark. The absorbance was measured at 595 nm. Ascorbic acid (0.01–1 mg/mL) was used as a positive control. The EC_50_ value was calculated using the curve of the percentage of ferric reducing antioxidant capacity corresponding with the various concentration of samples.

### 2.14. Statistical Analysis

All experiments were performed in triplicate (*n* = 3). The values are presented as means ± standard error of the mean (SEM). Data were analyzed using one-way analysis of variance (ANOVA), followed by Duncan’s post hoc test (SPSS version 17, SPSS Inc., Chicago, IL, USA). *p* < 0.05 was considered statistically significant.

## 3. Results

### 3.1. Total Phenolic and Anthocyanin Content

The total phenolic content of ABE was 300.9 ± 1.6 mg GAE/g extract, whereas the total anthocyanin content was 66.9 ± 1.3 mg cyanidin-3-glucoside/g extract.

### 3.2. LC-MS/MS Identification and Characterization of Phytochemical Compounds in ABE

The MS/MS spectral data of the phenolic composition of ABE in negative and positive modes are presented in Figure 1. The characteristics of ABE phenolic compounds were identified by comparing their retention time and mass spectral data of product ions with the online database (Table 1). As shown in Table 1, the proposed phytochemical compounds in ABE was quinic acid, gallic acid, 6-galloylglucose, 2,5-dihydroxybenzoic acid, ellagic acid, myricetin-3-galactoside, quercetin-3-O-arabinoside, quercetin-3-galactoside, kaempferol-3-rhamnoside, kaempferol-3-glucoside, delphinidin-3-glucoside, cyanidin-3-sambubioside, and cyanidin-3-glucoside.

### 3.3. Quantification of Individual Anthocyanins in ABE

The base peak and MS/MS fraction ion chromatogram of individual anthocyanins was obtained from LC-MS/MS analysis. D3G and C3G in ABE (Figure 2b) were characterized by comparing the retention time and mass spectral data of product ions with the standard (Figure 2a). D3G and C3G contents were quantified by comparing the calibration curve (Figure 3). The content of D3G and C3G in ABE was 21.7 ± 0.8 mg/g extract and 31.4 ± 1.4 mg/g extract, respectively.

### 3.4. The Inhibitory Effect of ABE on Carbohydrate Digestive Enzymes

The results showed that ABE (0.16–2.5 mg/mL) inhibited intestinal maltase (5.5–91.8%). Meanwhile, ABE (0.3–5 mg/mL) exhibited inhibitory activity against intestinal sucrase (6.5–97%). The IC_50_ values of ABE for intestinal maltase and sucrase were 0.76 ± 0.02 mg/mL and 1.33 ± 0.03 mg/mL, respectively (Table 2), indicating that ABE had greater inhibitory activity on maltase than sucrase. Moreover, acarbose (0.2–5 µg/mL) inhibited intestinal maltase by 21.2–75.8%, whereas at higher levels (1.6–50 µg/mL), it markedly inhibited intestinal sucrase by 16–85.1%. In addition, the IC_50_ values of acarbose against maltase and sucrase were 0.89 ± 0.07 µg/mL and 8.48 ± 0.07 µg/mL, respectively. The obtained results indicate that ABE has less potent than acarbose on the inhibition of carbohydrate digestive enzymes. In contrast, ABE (1–4 mg/mL) slightly inhibited the activity of pancreatic α-amylase by 11.2–34.1%, and the IC_50_ value was >4 mg/mL. Meanwhile, the IC_50_ value of acarbose was 34.62 ± 0.68 µg/mL.

### 3.5. The Inhibitory Activity on Protein Glycation

As shown in Figure 4, the formation of fluorescence AGEs in fructose and glucose-glycated BSA was 45.4-fold and 20.5-fold greater than that of BSA alone after four weeks of incubation (Figure 4A,B). In the fructose model, ABE (0.25 mg/mL) reduced the formation of AGEs, ranging from 48.9–58% throughout the four weeks of incubation. In the glucose model, the formation of AGEs was decreased by ABE 0.25 mg/mL; about 25.3–34.5%. At the same concentration, aminoguanidine (0.25 mg/mL) inhibited the formation of fluorescent AGEs in the fructose-glycated BSA (47.3%) and glucose-glycated BSA (52.6%). Our findings indicate that ABE (0.25 mg/mL) exhibits a similar potent as aminoguanidine (0.25 mg/mL) to inhibit fructose-induced BSA glycation.

The formation of N^ε^-CML in both BSA-fructose and BSA-glucose systems is shown in Figure 5. In the fructose and glucose model, the N^ε^-CML level was 6.6-fold and 1.4-fold higher than that of BSA alone at week four. Furthermore, the addition of ABE (0.25 mg/mL) and aminoguanidine (0.25 mg/mL) decreased the level of N^ε^-CML in fructose-glycated BSA by 35.0% and 28.2%, respectively. Whereas ABE (0.25 mg/mL) and aminoguanidine (0.25 mg/mL) inhibited the N^ε^-CML formation in glucose-glycated BSA by 26.2% and 35.5%, respectively. These findings indicate that the inhibitory effect of ABE (0.25 mg/mL) on N^ε^-CML level had a similar potency with aminoguanidine (0.25 mg/mL) in fructose and glucose models.

### 3.6. The Inhibitory Effect of ABE on Protein Oxidation

The protein carbonyl contents of fructose-glycated BSA are presented in Figure 6a. Our findings revealed that fructose increased the carbonyl contents by 4.9-fold to 11.8-fold as compared to non-glycated BSA during four weeks of incubation. The carbonyl content of fructose-glycated BSA was reduced by ABE (0.25 mg/mL) (24.8–38.8%) at weeks two and four, respectively. Moreover, ABE at 0.1 mg/mL decreased the carbonyl content of fructose-glycated BSA (32.2%) at week four. Whereas aminoguanidine (0.25 mg/mL) reduced the carbonyl content of fructose-glycated BSA by 34.4% (Figure 5a). As shown in Figure 6b, the glucose-mediated BSA glycation induced the carbonyl content higher than BSA throughout the four-week experiment (3.5-fold to 9.9-fold). The addition of ABE (0.1–0.25 mg/mL) significantly decreased the carbonyl content of glycated BSA at week three (15.5–26.7%) and week four (11.8–17.5%). Moreover, aminoguanidine (0.25 mg/mL) reduced glycated-glucose BSA’s carbonyl content by 29.6% (Figure 6b) at week four of incubation. These results indicate that ABE had a similar potency as aminoguanidine to decrease carbonyl content in fructose-glycated BSA. However, the ABE was less potent than aminoguanidine on reducing protein carbonyl content in BSA incubated with glucose.

### 3.7. The Inhibitory Effect of ABE on Protein Aggregation

At four weeks of the experimental period, the formation of β-amyloid cross structure, a marker of protein aggregation, was found to increase in glycated BSA induced by fructose (1.7-fold) and glucose (1.0-fold) when compared to BSA alone (Figure 7). Interestingly, ABE at a concentration of 0.25 mg/mL decreased the formation of β-amyloid in fructose- and glucose-glycated BSA (Figure 7A,B) about 38.5% and 17.9%, respectively. Similarly, the addition of aminoguanidine (0.25 mg/mL) caused a decrease in the formation of β-amyloid cross structure in glycated BSA induced by fructose (23.1%) and glucose (4.3%). According to these results, ABE was greater potent than aminoguanidine on the reduction of β-amyloid cross structure in glycated BSA.

### 3.8. Antioxidant Activity of ABE

As shown in Table 3, ABE had DPPH radical scavenging activity with an IC_50_ value of 15.84 ± 0.06 µg/mL, whereas the IC_50_ value of ascorbic acid was 11.83 ± 2.78 µg/mL. This finding indicated that ABE had less potency than ascorbic acid by 1.3-times. In the TEAC assay, the IC_50_ value of ABE was 1.3-fold higher than Trolox. In FRAP assay, ABE and ascorbic acid exhibited the EC_50_ value of 182.22 ± 0.64 µg/mL and 79.39 ± 1.92 µg/mL, respectively. The result suggested that ABE had 2.3-times less potent than ascorbic acid.

## 4. Discussion

Phytochemicals, especially anthocyanins, have been interesting to prevent or delay the progression of diabetes and its complications [36]. A meta-analysis of prospective cohort studies described that dietary anthocyanins and berry fruits had been associated with the reduced risk of type 2 diabetes risk [37]. One of the well-established antidiabetic mechanisms of anthocyanin-rich berries is the inhibition of pancreatic α-amylase, intestinal α-glucosidase, and glucose absorption [36,37,38,39]. The present findings exhibit the inhibitory activity of ABE against intestinal α-glucosidase, including maltase and sucrase in vitro. In addition, cyanidin-3-glucoside (C3G) and delphinidin-3-glucoside (D3G) are the major anthocyanins identified in ABE, which are agreed with a previous report in the methanolic extraction of *Antidesma bunius* [24]. Previous studies have suggested that anthocyanins could inhibit α-glucosidase activity. For example, the freeze-dried black currant extract containing anthocyanins, including 232 mg/kg C3G and 392 mg/kg D3G exhibited yeast α-glucosidase inhibition. In contrast, the green currant extract did not show an inhibitory effect on this enzyme [40]. Moreover, the study reported that the individual C3G and D3G at the concentration of 66 µg/mL also inhibited the α-glucosidase activity [40]. Additionally, C3G able to inhibit intestinal sucrase activity, and it showed the synergistic effect with acarbose on intestinal maltase and sucrase activities [41]. Similarly, D3G (100 µM) exhibited the inhibition of α-glucosidase by 44.5% [42]. Based on a previous study, the delphinidin-3-glucoside showed the most potency of α-glucosidase inhibition and was followed by cyanidin-3-glucoside, cyanidin-3-rutinoside, and malvidin-3-glucoside [43]. A previous study reported that anthocyanins could inhibit the intestinal α-glucosidase by the competitive inhibition between the glycosyl groups of anthocyanins and the active site of enzymes [44]. The scientific evidence demonstrates that cyanidin and delphinidin’s inhibitory effect on intestinal α-glucosidase may involve the hydrogen bond between hydroxyl groups of their structure and the polar groups in the active site of the enzyme [41]. There also found that the number of hydroxyl groups in the B-ring of anthocyanins associated with an increase of binding affinity to the enzyme [45]. Moreover, the inhibitory effect of C3G and D3G on α-glucosidase also depends on the binding affinity with the enzyme and their chemical structure [43]. According to the molecular docking study, the binding of D3G and α-glucosidase depends on the polar interaction, hydrogen bonding, and hydrophobic interaction in the enzyme’s catalytic site [42]. It is noteworthy that the molecular docking predicted D3G could bind with amino acid residues (Gly402 and Val380) of α-glucosidase through hydrogen bonding. In the meantime, C3G showed the binding affinity with amino residues, including Glu231, Val335, Gly402, and Val380 [43]. Therefore, it is hypothesized that ABE could inhibit α-glucosidase (maltase and sucrase) through the interaction of enzymes resulting from cyanidin-3-glucoside and delphinidin-3-glucoside as a predominant anthocyanin in ABE.

Chronic hyperglycemia spontaneously modifies circulating and structural proteins through induction of protein glycation, leading to advanced glycation end productions (AGEs). In general, AGEs can be categorized by their chemical structures into two main groups: fluorescent and cross-link AGEs (e.g., pentosidine, crossline, and methyl glyoxal-lysine dimer or MOLD) and non-fluorescent and non-crosslink AGEs (e.g., N^ε^-(carboxymethyllysine) or CML, N^ε^-(carboxyethyllysine) or CEL, and pyrraline) [46]. In the present study, fructose has a higher protein glycation rate than glucose because ketose sugar has more reactive than aldose sugar [47,48]. Moreover, fructose has a more open-chain structure than glucose, resulting in a higher reactivity than glucose [49]. ABE (0.25 mg/mL) decreased the formation of fluorescent AGEs and the level of N^ε^-CML in fructose and glucose-glycated BSA throughout the study period. Furthermore, ABE (0.25 mg/mL) exhibited a similar effect on the reduction of AGE formation as aminoguanidine (0.25 mg/mL) in the fructose system. According to a previous study, C3G prevented the lysine and arginine residues of the protein, leading to reduced fructosamine formation and BSA glycation [50]. Interestingly, the molecular docking indicated that C3G exhibited BSA glycation’s competitive inhibition through the hydrogen bonding with Glu186, Arg427, Ser428, Lys431, Arg435, and Arg458. Moreover, C3G also reduced the BSA glycation by hydrophobic interaction with Leu189 and Ile455 [50].

During the protein glycation, the monosaccharide-mediated BSA glycation induced the change of helical structure of BSA to the β-sheet structure, resulting in the formation of insoluble proteins, known as amyloids [50,51]. The previous study has reported that the protein glycation decreased the secondary α-helix structure of BSA and increased the β-sheet structure formation [52]. The formation of the cross-β structure may involve two pathways, including (i) the covalent bonding between carbohydrate and amino acid residues (Lys and Arg) leading to unfolding protein, (ii) the intramolecular or intermolecular AGE-crosslinks resulting in the local or global unfolding of protein [53]. Consequently, the unfold polypeptides rearrange to form the amyloid fibrils [53]. Our findings demonstrated that fructose-induced the formation of the β-amyloid cross structure higher than glucose, whereas ABE could reduce the β-amyloid cross structure level. A previous study revealed that C3G could decrease amyloid fibril level during the protein glycation [50]. According to the molecular docking study, C3G entered the hydrophobic cavity in the subdomain IIA of BSA. Moreover, C3G interacted with BSA through Van der Waals forces and hydrogen bonding that could prevent BSA’s secondary structure [54]. In addition, the inhibitory effect of anthocyanin glycoside on amyloid formation is involved in the interaction between the aromatic ring of anthocyanins [55]. Therefore, it could be hypothesized that ABE may help preserve BSA structure, resulting in reduced fructose- and glucose-induced AGE and β-amyloid formation. The decreasing of the β-amyloid cross structure is helpful to prevent or delay the progression of diabetic complications.

The reactive oxygen species (ROS) such as superoxide anion and hydrogen peroxide are generated via the oxidation of Schiff bases or Amadori products during the early stage of the glycation process [56,57]. Moreover, the interaction of AGEs and its receptor also modulates ROS production through the MAPK pathway [58]. The ROS oxidize the thiol group of amino acid cysteine and methionine in protein molecules, resulting in the loss of thiol groups and enzyme inactivity [59,60]. In addition, the oxidation of amino acids (Arg, Lys, and Thr) results in increasing protein carbonyl content [61]. Our findings found that the fructose and glucose-mediated BSA glycation continuously increased protein carbonyl levels throughout the experimental period. Previous studies showed that edible plants containing phenolic compounds reduced the protein oxidation in glycated BSA. For example, *Mesona chinesis* extract (0.5–1 mg/mL) decreased the protein carbonyl content in fructose-mediated BSA glycation by 36.2–46.7% [62]. The flower petal extract of *Clitoria ternatea* containing anthocyanins and phenolic compounds prevented the formation of protein carbonyl by 8.23–11.34% at the concentration of 0.25–1 mg/mL [23]. In the present study, ABE reduced the level of protein carbonyl content in glycated BSA during the four weeks of the experimental period. Several studies support that the free radical scavenging activity of flavonoids may be the mechanism for glycation inhibition and the reduction of protein carbonyl content [33,34,47]. Flavonoids could scavenge free radicals produced from the interaction between reducing sugars and protein during the Schiff bases oxidation, leading to blocking the formation of protein glycation and oxidation [20].

In the current study, ABE demonstrated antioxidant activity, indicated by DPPH scavenging activity, TEAC, and FRAP. A previous study described that 10 µM of C3G or D3G scavenged DPPH radicals with the percentage of inhibition of 32% and 42%, respectively [63]. The scientific evidence demonstrates that the hydroxylation and methoxylation in the B ring of their structure impact the ability of C3G and D3G for DPPH radicals scavenging. The increased hydroxyl groups in the B ring could increase the radical scavenging activity indicating that delphinidin decreased free radicals greater than cyanidin [64]. Additionally, the highest value of ferric reducing ion power among anthocyanins was found in cyanidin-3-glucoside and delphinidin-3-glucose, following petunidin-3-glucoside, peonidin-3-glucoside, and malvidin-3-glucoside. The reducing capacity is influenced by the pyrogallol or catechol type on B rings of their structure [65]. According to those mentioned above, it is hypothesized that anthocyanins in *Antidesma bunius* may be responsible for the inhibitory activity against protein glycation and the prevention of oxidative damage to BSA.

## 5. Conclusions

ABE exhibits antioxidant activity and inhibits carbohydrate digestive enzymes, including intestinal maltase and sucrase. In the glycation model, ABE also inhibits the reduction of fluorescent AGE formation, N^ε^-CML level, protein carbonyl content, and β-amyloid formation. ABE may be a promising ingredient that helps suppress carbohydrate digestion and prevent monosaccharide-mediated protein glycation, oxidation, and aggregation.

## Figures and Tables

**Figure 1 antioxidants-10-00032-f001:**
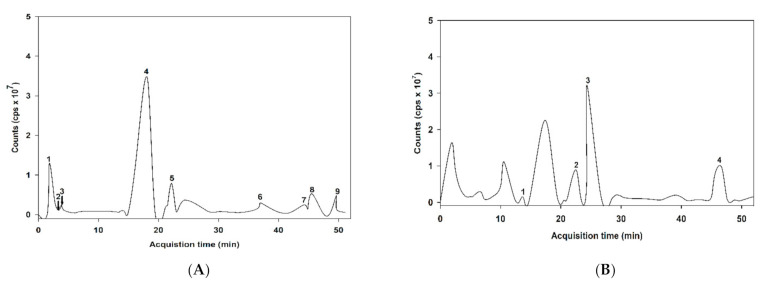
liquid chromatography-mass spectrometry (LC-MS/MS) chromatogram of the proposed phenolic compound of anthocyanin-rich fraction in *Antidesma bunius* fruit extract (ABE): (**A**) negative of ionization mode; (**B**) positive of ionization mode.

**Figure 2 antioxidants-10-00032-f002:**
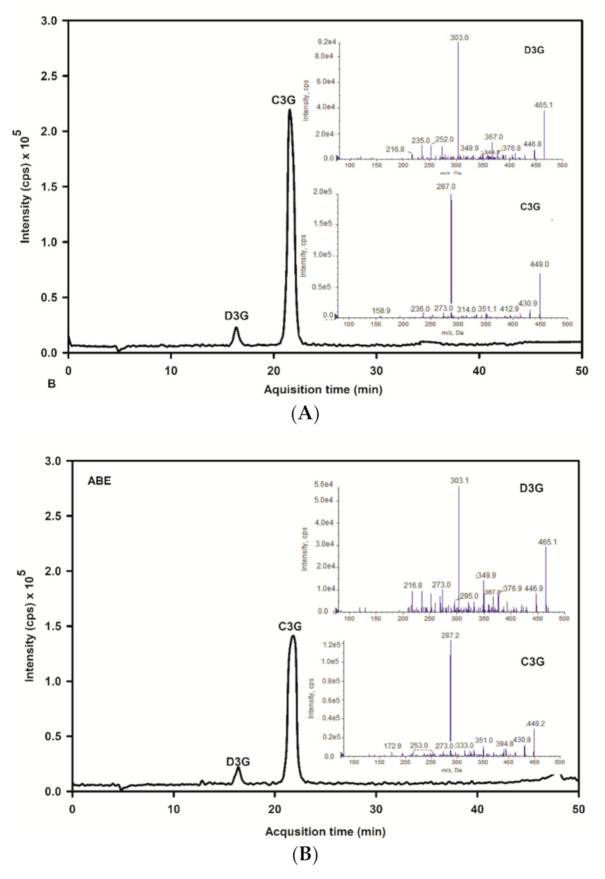
LC-MS/MS chromatogram of delphinidin-3-glucoside and cyanidin-3-glucoside of the anthocyanin-rich fraction of *Antidesma bunius* extract (ABE) with multiple reaction monitoring-enhanced product ion mode (MRM-EPI): (**A**) chromatogram of anthocyanins standard; D3G: delphinidin-3-glucoside and peak and C3G: cyanidin-3-glucoside; (**B**) chromatogram of ABE; D3G: delphinidin-3-glucoside and C3G: cyanidin-3-glucoside.

**Figure 3 antioxidants-10-00032-f003:**
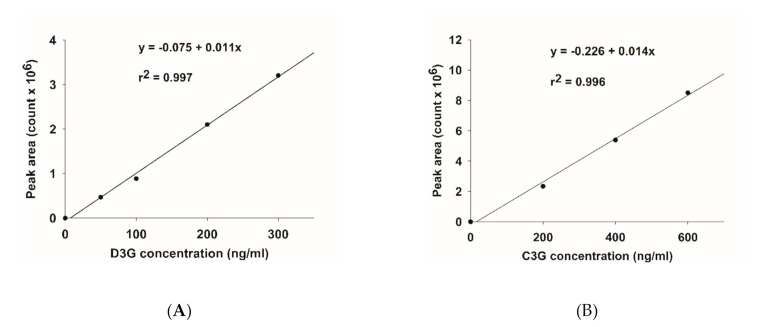
Calibration curve of anthocyanins: (**A**) delphinidin-3-glucoside (D3G); (**B**) cyanidin-3-glucoside (C3G).

**Figure 4 antioxidants-10-00032-f004:**
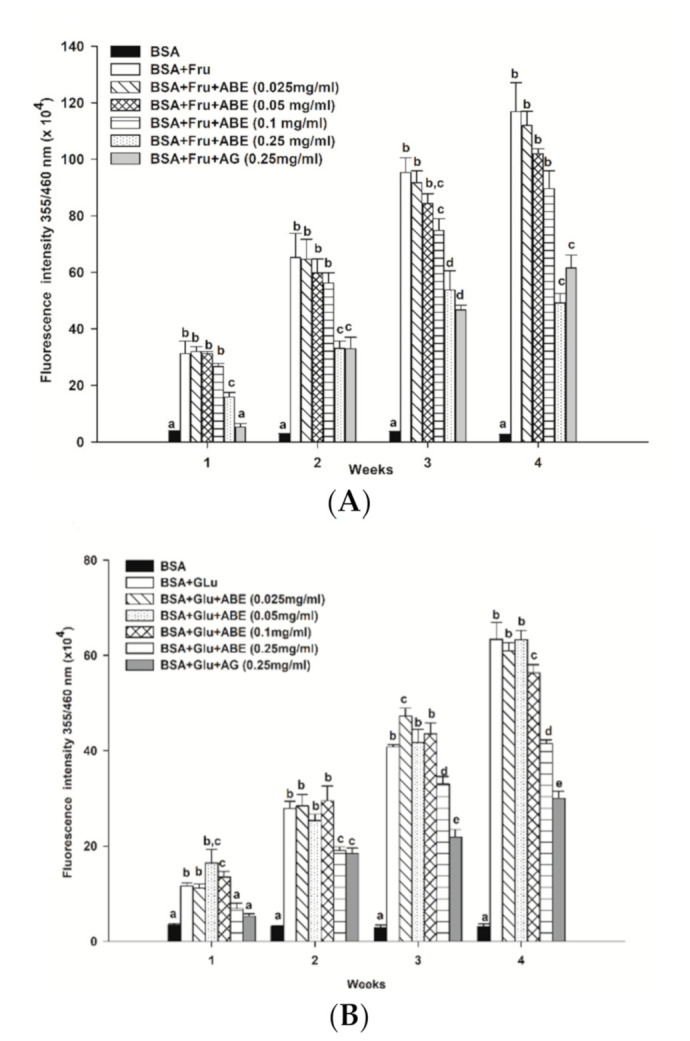
The effect of anthocyanin-rich fraction *Antidesma bunius* fruit extract (ABE, 0.025–0.25 mg/mL) and aminoguanidine (AG, 0.25 mg/mL) on the fluorescent AGE formation in bovine serum albumin (BSA) incubated with (**A**) 0.5 M fructose (Fru); (**B**) 0.5 M glucose (Glu) during 4 weeks of study. The results are expressed as mean ± SEM (*n* = 3). Results were statistically analyzed by one-way analysis of variance (ANOVA), following Duncan’s multiple range test. The different letters in the same week indicate a significant difference at *p* < 0.05.

**Figure 5 antioxidants-10-00032-f005:**
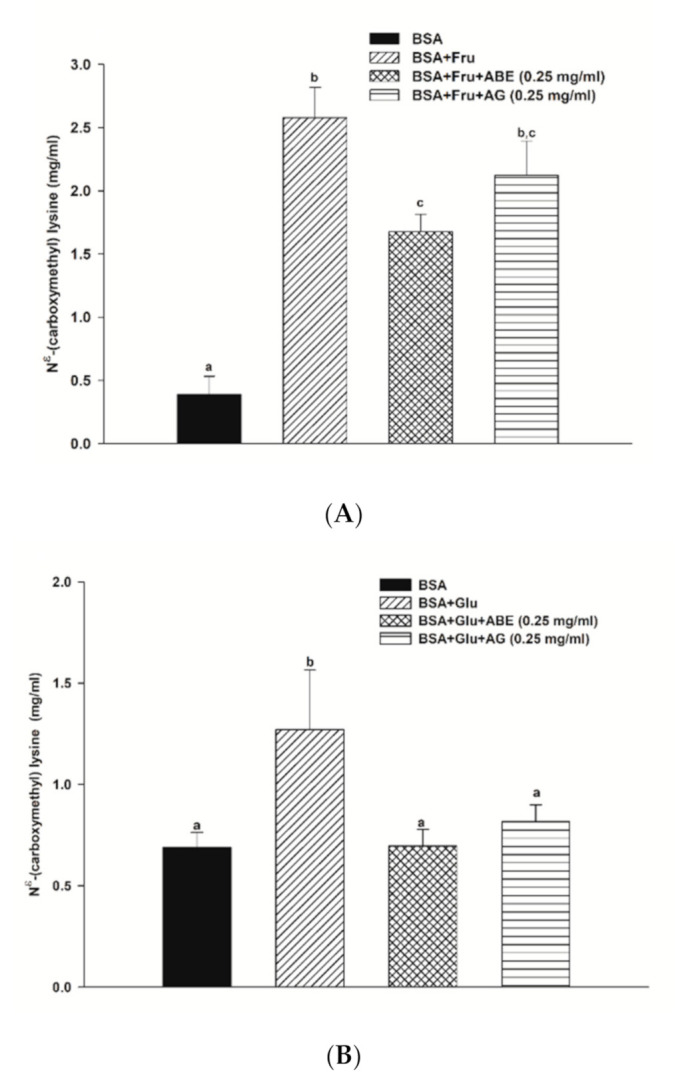
The effect of anthocyanin-rich fraction *Antidesma bunius* fruit extract (ABE, 0.25 mg/mL) and aminoguanidine (AG, 0.25 mg/mL) on the N^ε^-(carboxymethyl) lysine (N^ε^-CML) level in bovine serum albumin (BSA) incubated with (**A**) 0.5 M fructose (Fru); (**B**) 0.5 M glucose (Glu) during 4 weeks of study. The results are expressed as mean ± SEM (*n* = 3). Results were statistically analyzed by one-way analysis of variance (ANOVA), following Duncan’s multiple range test. The different letters indicate Scheme 0.

**Figure 6 antioxidants-10-00032-f006:**
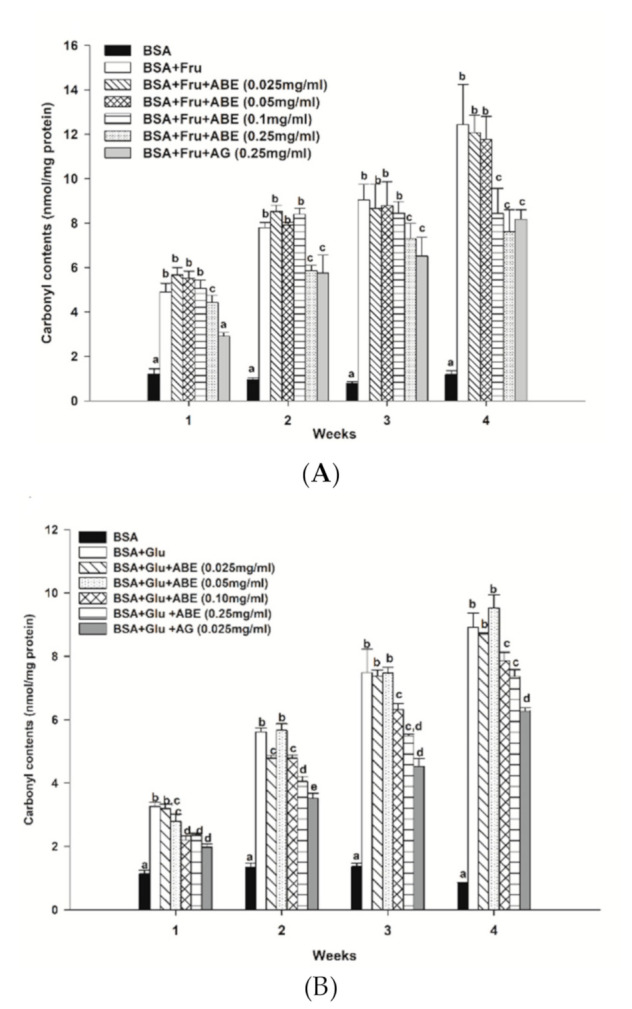
The effect of anthocyanin-rich fraction *Antidesma bunius* fruit extract (ABE, 0.025–0.25 mg/mL) and aminoguanidine (AG, 0.25 mg/mL) on the level of protein carbonyl content in bovine serum albumin (BSA) incubated with (**A**) 0.5 M fructose (Fru); (**B**) 0.5M glucose (Glu) during the four weeks of study. The results are expressed as mean ± SEM (*n* = 3). Results were statistically analyzed by one-way analysis of variance (ANOVA), following Duncan’s multiple range test. The different letters in the same week indicate a significant difference at *p* < 0.05.

**Figure 7 antioxidants-10-00032-f007:**
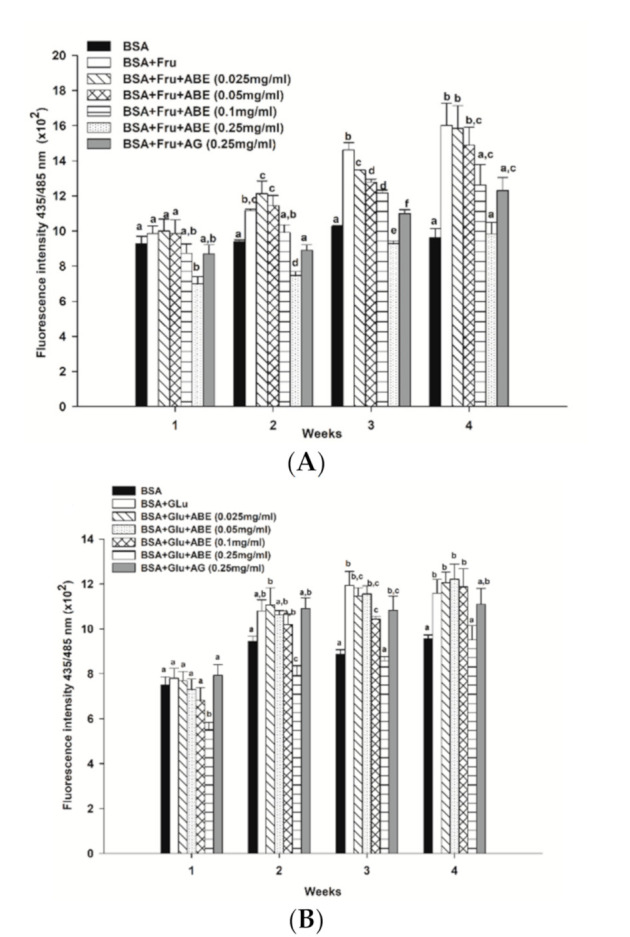
The effect of anthocyanin-rich fraction *Antidesma bunius* fruit extract (ABE, 0.025–0.25 mg/mL) and aminoguanidine (AG, 0.25 mg/mL) on the level of β-amyloid cross structure in bovine serum albumin (BSA) incubated with (**A**) 0.5 M fructose (Fru); (**B**) 0.5 M glucose (Glu) during the four weeks of study. The results are expressed as mean ± SEM (*n* = 3). Results were statistically analyzed by one-way analysis of variance (ANOVA), following Duncan’s multiple range test. The different letters in the same week indicate a significant difference at *p* < 0.05.

**Table 1 antioxidants-10-00032-t001:** Chromatographic mass spectrometry (MS) and MS/MS data of proposed phenolic compounds in an anthocyanin-rich fraction of *Antidesma bunius* fruit extract (ABE).

Peak No.	Rt (min)	Proposed Compounds	Product Ion
		**Phenolic acids**	
1	1.743	Quinic acid	192.0639 (M − H)^−^, Calcd Mass: 191.0567
2	3.228	Gallic acid	170.0217 (M − H)^−^, Calcd Mass 169.0144
3	4.076	6-Galloylglucose	332.0743 (M − H)^−^, Calcd Mass: 331.0671
4	21.945	2,5-dihydroxybenzoic acid	154.0266 (M − H)^−^, Calcd Mass: 153.0194
5	45.200	Ellagic acid	302.0063 (M − H)^−^, Calcd Mass: 300.999
		**Flavonols**	
6	18.244	Quercetin-3-O-arabinoglucoside	596.1379 (M − H)^−^, Calcd Mass: 595.1308, MS/MS: 300.0274
7	36.954	Quercetin-3-galactoside	464.0953 (M − H)^−^, Calcd Mass: 463.0883, MS/MS: 301.0345
8	44.893	Kaempferol -3-rhamnoside	432.1052 (M − H)^−^, Calcd Mass:431.098, MS/MS: 283.0605
9	49.541	Kaempferol-3-glucoside	448.1006 (M − H)^−^, Calcd Mass: 447.0936, MS/MS: 285.4000
10	13.427	Myricetin-3-galactoside	480.1576 (M − H)^+^, Calcd Mass: 481.1652, MS/MS: 319.1124
		**Anthocyanins**	
11	22.484	Delphinidin-3-glucoside	465.1811 (M)^+^, Calcd Mss: 465.1715, MS/MS: 303.1166
12	24.200	Cyanidin-3-sambubioside	581.138 (M)^+^, Calcd Mass: 581.1431, MS/MS: 287.0523
13	46.069	Cyanidin-3-glucoside	448.1714 (M − H)^+^, Calcd Mass: 449.1802, MS/MS: 287.1236

**Table 2 antioxidants-10-00032-t002:** The inhibitory activity of anthocyanin-rich fraction of *Antidesma bunius* fruit extract (ABE) on carbohydrate digestive enzymes.

Compounds	IC_50_ Values		
	α-Amylase	Maltase	Sucrase
**ABE (mg/mL)**	>4	0.76 ± 0.02	1.33 ± 0.03
**Acarbose (µg/mL)**	34.62 ± 0.68	0.89 ± 0.07	8.48 ± 0.07

The results are expressed as mean ± SEM (*n* = 3). IC_50_: Half-maximal inhibitory concentration.

**Table 3 antioxidants-10-00032-t003:** The antioxidant capacities of the anthocyanin-rich fraction of *Antidesma bunius* fruit extract (ABE).

Compounds	DPPH IC_50_ Value (µg/mL)	TEAC IC_50_ Value (µg/mL)	FRAP EC_50_ Value (µg/mL)
**ABE**	15.84 ± 0.06	166.10 ± 2.40	182.22 ± 0.64
**Ascorbic acid**	11.83 ± 2.78	-	79.39 ± 1.92
**Trolox**	-	215.36 ± 0.13	-

DPPH: 2,2-diphenyl-1-picrylhydrazyl; TEAC: Trolox equivalent antioxidant capacity; FRAP: ferric reducing antioxidant power; IC_50_: Half-maximal inhibitory concentration; EC_50_: Half-maximal effective concentration.

## Data Availability

Data is contained within the article.

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
