# Peer review of "Inhibitory Effect of Antidesma bunius Fruit Extract on Carbohydrate Digestive Enzymes Activity and Protein Glycation In Vitro"

_antioxidants, 2020, doi:10.3390/antiox10010032_

Round 1
Reviewer 1 Report
I think the article is interesting and well presented. I have minor suggestions:
1.I proposes to remove Fig. 2, because the data are given in the text and they are difficult to read due to small letters used.
2. Authors should state in the text if the ABE extract could inhibit lactase activity or not.
3. In order to be clearly referred, the authors should suply the doi number of the references, 16, 19, 22, 23, 26, 34, 40, 43, 45, 51, and 62
Author Response
Point 1: I proposes to remove Fig. 2, because the data are given in the text and they are difficult to read due to small letters used.
Response 1: Cyanidin-3-glucoside (C3G) and delphinidin-3-glucoside (D3G) are major anthocyanins of ABE. Could I supply Figure 2A-B with large size?
Point 2: Authors should state in the text if the ABE extract could inhibit lactase activity or not.
Response 2: According to a previous study, a mulberry extract containing anthocyanins exhibited the inhibition of lactase activity in vitro study. It is interesting to investigate the inhibitory effect of ABE extract on lactase activity. However, the present study focused on the inhibitory effect of ABE on maltase and sucrase.
Point 3: In order to be clearly referred, the authors should supply the doi number of the references, 16, 19, 22, 23, 26, 34, 40, 43, 45, 51, and 62
Response 3: We supplied the doi number of the references 16, 19, 22, 23, 26, 34, 40, 43, 45, 51, and 62.
Reviewer 2 Report
This manuscript reports the inhibitory activity of the anthocyanin-rich fraction of Antidesma bunius fruit extract (ABE) on pancreatic α-amylase and intestinal α-glucosidase, the inhibitory effect on fructose/glucose-induced protein glycation, oxidative protein damage, and protein aggregation, was well as the phytochemical composition and antioxidant activity. This subject is worth of investigation and is in line with the scope of Antioxidants, and the presented work is well-structured, well-written and easy to understand.
Some comments and/or suggestions:
- The authors could provide the calibration curves (and coefficient of determination) constructed with the standard compounds used in the quantification of anthocyanins.
- Line 172. Indicate the ABE concentrations used in the assay.
- Line 189. Indicate the concentration of the positive control ascarbose.
- Line 237. Indicate the concentration of the positive control trolox.
- Line 245. Indicate the concentration of the positive control ascorbic acid.
- Line 275. “quantified by comparing the standard” … this sentence should be rewritten to clarify how the quantification was done.
- Figure 2 is difficult to visualize. Perhaps it can be supplied with a larger size.
Author Response
Point 1: The authors could provide the calibration curves (and coefficient of determination) constructed with the standard compounds used in the quantification of anthocyanins.
Response 1: We provided the calibration curve of delphinidin-3-glucoside (D3G) and cyanidin-3-glucoside with the coefficient, as shown in Figure 3.
Point 2: Line 172. Indicate the ABE concentrations used in the assay.
Response 2: The ABE concentration (15-60 mg/ml) was indicated in line 172 for the inhibition of carbohydrate digestive enzymes assay.
Point 3: Line 189. Indicate the concentration of the positive control acarbose.
Response 3: The concentration of acarbose (0.16-5 µg/ml) was indicated in line 189 for the inhibition of carbohydrate digestive enzymes assay
Point 4: Line 237. Indicate the concentration of the positive control trolox.
Response 4: The concentration of trolox (0.06-1 mg/ml) was indicated in line 237 for trolox equivalent antioxidant capacity assay.
Point 5: Line 245. Indicate the concentration of the positive control ascorbic acid.
Response 5: The concentration of ascorbic acid (0.01-1 mg/ml) was indicated in line 245 for ferric reducing antioxidant power assay.
Point 6: “quantified by comparing the standard” … this sentence should be rewritten to clarify how the quantification was done.
Response 6: We rewrote these sentences "D3G and C3G in ABE (Figure 2b) were characterized by comparing the retention time and mass spectral data of product ions with the standard (Figure 2a). D3G and C3G contents were quantified by comparing the calibration curve (Figure 3)."
Point 7: Figure 2 is difficult to visualize. Perhaps it can be supplied with a larger size.
Response 7: We supplied with a larger size, as shown in Figure 3.
Reviewer 3 Report
Overall, the submitted manuscript is well-structured and the multidirectional approach to evaluate antidiabetic activity
The HPLC-MS analysis is a fundamental cornerstone for the investigation of plant secondary metabolites. the approach followed by authors is appreciable. However, some concerns arise from the peak resolution of samples, as depicted in figures 1A-B. I suggest to repeat the chromatographic runs in order to ameliorate the separation.
Additionally, it is not provided the scan mass range.
The statistical analysis description in figures 3-6 should be ameliorated. Specifically, the ANOVA P values should be included alongside with the Pvalues related to post hoc test.
Author Response
Point 1:The HPLC-MS analysis is a fundamental cornerstone for the investigation of plant secondary metabolites. the approach followed by authors is appreciable. However, some concerns arise from the peak resolution of samples, as depicted in figures 1A-B. I suggest to repeat the chromatographic runs in order to ameliorate the separation.
Response 1: We supplied the LC-MS/MS chromatogram of ABE extract with negative ionization mode and positive ionization mode that ameliorated the peak separation, as shown in Figure 1A-B.
Point 2: The statistical analysis description in figures 3-6 should be ameliorated. Specifically, the ANOVA P values should be included alongside with the Pvalues related to post hoc test.
Response 2: We indicated P-values related to post hoc test with the sentences "Results were statistically analyzed by one-way ANOVA, following Duncan’s multiple range test. The different letters indicate a significant difference at p<0.05." (Figure 4-7)